# GROUNDING EVERYTHING:
# EMERGING LOCALIZATION PROPERTIES
# IN VISION-LANGUAGE TRANSFORMERS

## ABSTRACT

Vision-language models have shown remarkable performance in various fields, ranging from zero-shot classification to captioning and prompt-based image generation. But so far, those models do not seem able to localize referential expressions and objects in images, with the result that they are only used as a post-process labeling step or that they need to be fine-tuned for this task. The following work, we show that vision-language (VL) models trained with image-level objectives hold object localization properties. We propose a Grounding Everything Model (GEM) that allows to leverage these properties without retraining or fine-tuning the pretrained model. To this end, we extend the idea of v-v attention introduced by CLIPSurgery (Li et al., 2023b) to a generalized self-self attention path and propose a set of regularizations that allows the model to better generalize across datasets and backbones. We further show how the concept of self-self attention corresponds to clustering, thus enforcing groups of tokens arising from the same object to be similar while preserving the alignment with the language space. We evaluate the proposed GEM framework on three benchmark datasets and improve the performance in training-free open-vocabulary localization.

## 1 INTRODUCTION

The recent availability of web-scaled datasets (Schuhmann et al., 2022; Gadre et al., 2023) fostered significant progress in vision-language models, enabling training on vast image-text pair datasets. These models (Radford et al., 2021; Jia et al., 2021; Li et al., 2022a; 2019), exhibit the ability to generalize to various downstream tasks like zero-shot image classification (Radford et al., 2021; Jia et al., 2021; Cherti et al., 2023), visual question answering (Khan et al., 2022), action recognition (Yuan et al., 2021; Yu et al., 2022), image captioning (Li et al., 2022a; 2019), and view synthesis (Jain et al., 2021). However, models trained with image-level objectives such as contrastive loss, image-text matching, or image captioning struggle to maintain their zero-shot capabilities for tasks requiring visual localization. In order to leverage VL models to localize objects in an open-vocabulary setting, different streams of approaches have been proposed. The first line of work uses pretrained VL models for post-process labeling after the localization resp. segmentation such as RegionCLIP (Zhong et al., 2022) or Semantic-SAM (Li et al., 2023a)). In this case, the localization process is independent of the actual referential expression. A second line of work retrains the VL models to improve localization, e.g. PACL (Mukhoti et al., 2023) or GroupViT Xu et al. (2022). In contrast to that, a third group recently started to explore the inherent localization capabilities of models trained on global objectives without the need for architectural modifications or retraining, namely MaskCLIP (Zhou et al., 2022), MaskCLIP[(2)] (Dong et al., 2023), and CLIPSurgery (Li et al., 2023b). In particular, CLIPSurgery extends the pretrained ViT backbone of the CLIP model by a so-called "surgery pathway" which accumulates the value-value attentions of the original backbone over several layers. This tends to produce spatial groups of tokens related to individual objects in the last layer, which are also aligned to the vision language space. While the introduction of the surgery pathway shows a significant performance increase compared to the original CLIP performance, it is not clear how this mechanism impacts the overall processing to achieve the respective results. In this paper, we analyze the properties of VL transformer models that result in the characteristics observed for CLIP surgery and propose to enforce them within a generalized self-self attention architecture.

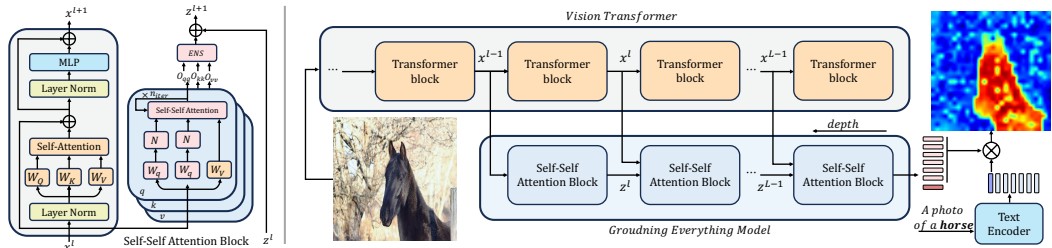

Figure 1: **Grounding Everything Model architecture:** (left) Illustration of the proposed generalized self-self attention block including iteration and $L^2$ normalization $N$. The output of each q-q, k-k, and v-v projection is ensembled before applying the skip connection. (right) The output of self-self attention blocks is aggregated in parallel to the vision transformer in an alternative pathway. The segmentation prediction is obtained by the dot product between the patch token output of the GEM and the text embedding of the text encoder.

First, we show that the v-v attention proposed by Li et al. (2023b) can be generalized to a self-self attention, as any k-k, q-q, or v-v representations show similar characteristics. Practically, it shows that any form of self-self attention increases similarity among groups of similar tokens, compared to the standard q-k attention. To control the process of group formation, we propose a set of regularizations, namely the $L^2$ normalization of the projected vectors, the iteration over self-self attention, as well as an adaptive temperature. Finally, we ensemble over all possible self-self attention types to allow for an integration of all cues. An overview of the resulting Grounding Everything Model (GEM) architecture is given in Figure 1.

We evaluate the proposed method on three challenging downstream datasets, PascalVOC (Everingham et al., 2010), PascalContext (Mottaghi et al., 2014), as well as on the large-scale OpenImages V7 (Benenson & Ferrari, 2022). In all cases, we show improved results over previous training-free methods (Li et al., 2023b; Zhou et al., 2022) and even competitive results in comparison to other approaches that require some form of fine-tuning (Xu et al., 2022; 2023; Luo et al., 2023). We further evaluate the method on different ViT architectures (Dosovitskiy et al., 2021), showing its generalizability across different depths, as well as on various publicly available VL models beyond CLIP, such as OpenCLIP (Cherti et al., 2023), and BLIP (Li et al., 2022a).

We summarize our contributions as follows: (1) Inspired by Li et al. (2023b), we propose a self-self attention pipeline for open-vocabulary referential expression localization and segmentation based on pretrained vision-language models. (2) We propose the Grounding Everything Model (GEM) as a combination of self-self attention together with a set of regularizations that allows to generalize over a range of VL models and datasets. (3) We provide an in-depth analysis of the properties of self-self attention and how they lead to an improved localization in pretrained ViT transformer models.

## 2 RELATED WORKS

The success of large-scale VL models like CLIP has sparked the interest to leverage their abilities for tasks like open-vocabulary object localization. Given the lack of localization properties of VL models, one line of approaches uses them as a form of post-process labeling after an initial localization step. For example, RegionCLIP (Zhong et al., 2022) uses a supervised bounding box proposal (Ren et al., 2015) to localize objects and utilizes CLIP to classify them. Similarly, OpenSeg (Ghiasi et al., 2022) leverages ALIGN (Jia et al., 2021) by fine-tuning it using class-agnostic masks and image-text pair data, while NamedMask (Shin et al., 2023) relies on an unsupervised saliency detector for the localization. By relying on an external model not trained on a web-scale dataset with a large vocabulary, the localization performance is capped by the quality of that model. Another line of work leverages various sources of region-level supervision such as masks or bounding boxes available for different vision tasks. For example, X-Decoder (Zou et al., 2023a) uses all types of image segmentation annotation (semantic, instance, and panoptic), as well as curated datasets for VQA and image captioning. SEEM (Zou et al., 2023b) further includes interactive segmentation. GLIP (Li et al., 2022b; Zhang et al., 2022) is trained based on a combination of object detection and grounding datasets. Other examples include Grounding DINO (Liu et al., 2023), Semantic-SAM (Li et al., 2023a) and OpenSeeD (Zhang et al., 2023). Combining the supervision from various tasks

allows these models to be trained on millions of samples with fine-grained supervision and thus achieving high performance for a large set of tasks. However, relying on region-level annotated data makes this approach difficult to scale and limits its usage in domains where such annotations are expensive to obtain (Blumenstiel et al., 2023).

Alternatively, some works propose to adapt the VL model architecture and training process to favor the emergence of localization. SegCLIP (Luo et al., 2023) and GroupViT (Xu et al., 2022) modify the ViT architecture by interleaving regular transformer blocks with grouping blocks that allow the grouping of semantically similar tokens into learnable group tokens used to compute the contrastive loss with the text. Similarly, ViL-Seg (Liu et al., 2022) and OVSegmentor (Xu et al., 2023) respectively use online clustering and Slot Attention (Locatello et al., 2020) for grouping visual features into semantically coherent clusters and in addition exploit self-supervision for refinement. Alternatively, ReCo (Liu et al., 2021) leverages a retrieval process to obtain finer supervision and PACL (Mukhoti et al., 2023) trains a decoder on top of CLIP with a grounding loss. While these methods use image-caption pairs as supervision, they require heavy filtering of the dataset, like extracting common nouns, which makes the dataset lose its free-form text characteristic. Thus, such approaches do not fully benefit from the VL models' large-scale characteristics.

Some methods refrain from training and instead adopt the pretrained VL model to make them work on fine-grained localization tasks. MaskCLIP (Zhou et al., 2022) proposes to discard the last MLP of the vision transformer and use the last value projection to extract dense patch-level features. CLIPSurgery (Li et al., 2023b) builds upon that and proposes to introduce a new pathway, computed in parallel to the vision encoder. We discuss both methods in detail in the following section.

## 3 BACKGROUND: CLIP SURGERY

In this section, we review MaskCLIP (Zhou et al., 2022) and CLIP Surgery (CS) (Li et al., 2023b), two extensions of the CLIP vision-language model.MaskCLIP (Zhou et al., 2022) proposes discarding the last Multi-Layer Perceptron (MLP) of the vision transformer and to utilize the final value projection to extract dense patch-level features. Building upon this concept, CLIPSurgery (Li et al., 2023b) introduces a novel pathway called the "surgery pathway" that operates in parallel with the original vision encoder. The surgery pathway in CLIPSurgery is an extension of the original Vision Transformer (ViT) backbone of the CLIP model. It employs value-value attention defined as:

$$Attn_{vv} = \text{softmax}(V \cdot V^T), \quad O_{vv} = Attn_{vv} \cdot V \tag{1}$$

with $V = xW_v \in \mathbb{R}^{n \times d}$, with $x$ representing the patch tokens output by a ViT layer, $n$ represent the number and $d$ the dimension of tokens, respectively, and $W_v$ is the learned value weight matrix of the original ViT backbone, and $O_{vv}$ is the output of the value-value surgery block. The output of multiple layers is aggregated via residual connection, resulting in a second set of tokens. Note that the value-value attention is directly used without a subsequent MLP and can be aggregated over several layers. To localize an object based on an input label or referential expression, the distance is computed between the token output of the last layer and the respective text embedding.

## 4 GROUND EVERYTHING THROUGH SELF-SELF ATTENTION

In the following, we introduce the Ground Everything Model (GEM) by first generalizing the concept of value-value attention (Li et al., 2023b) to a broader set of projections as self-self-attention and introduce an iterative extension that, together with a temperature regularizer, allows to control the formation of groups of visual features. Second, we consider the connection of the proposed self-self attention (and also CLIPSurgery's value-value attention) to clustering, showing in simulations that it can act as a form of clustering.

### 4.1 GEM: GROUNDING EVERYTHING MODEL

**Self-Self Attention:** We first review the concept of value-value attention, showing that, while it allows connecting features from the same semantic region, the same properties can be observed for key-key or query-query projections. We verify this idea by replacing the value projection by either

the query or the key projection taken from the original pathway. Table 1 shows that the query-query and key-key attention leads to the same or improved performance. We, therefore, introduce a generalized self-self attention as extension of the value-value attention as:

$$Attn_{ss} = \text{softmax}(xW_{proj} \cdot (xW_{proj})^T), \quad O_{ss} = Attn_{ss} \cdot V \qquad (2)$$

with $x \in R^{n \times d}$ again representing the patch tokens output by a ViT layer, and $W_{proj}$ being a projection matrix of the respective ViT layer $W_{proj} \in \{W_v, W_q, W_k\}$. Compared to regular self-attention (qk-attention), self-self-attention increases the similarity of already similar patch tokens, thus leading to cluster formation (see Section 4.2).

**Normalization:** In the self-self attention setting, projected tokens with high norms might disproportionately influence other tokens, regardless of their similarity with other visual tokens. We therefore propose an $L^2$-normalization for each projected token before computing self-self attention. This normalization improves the group formation and thus the localization as shown in Table 1.

**Iterative Self-Self Attention:** We iteratively apply the proposed normalized self-self attention to facilitate the gradual refinement of the cluster formation of semantically related visual tokens. More formally, given input visual tokens denoted as $x \in \mathbb{R}^{n \times d}$ and a projection matrix $W_{proj} \in \mathbb{R}^{d \times d}$, the $k$-th iteration of our iterative-self-self attention can be described as:

$$\begin{cases} p^0 & = \frac{xW_{proj}}{||xW_{proj}||_2} \\ p^{k\prime} & = softmax(p^{k-1} \cdot (p^{k-1})^T) \cdot p^{k-1} \\ p^k & = \frac{p^{k\prime}}{||p^k||_2} \end{cases} \qquad (3)$$

After $K$ iterations of self-self attention, the output (for the $W_{proj}$ projection), denoted $O_{ss}$, is obtained by applying the assignment to the values since they are trained to carry semantic information:

$$O_{ss} = softmax(p^K \cdot (p^K)^T) \cdot V \qquad (4)$$

**Adaptive Temperature:** We can further guide the cluster formation by introducing a temperature $\tau$ in the softmax formulation as:

$$softmax_\tau(a)_{ij} = \frac{e^{a_i \cdot a_j^T / \tau}}{\sum_k e^{a_i, a_k^T / \tau}} \qquad (5)$$

Assuming a zero-shot setting without access to labeled training or validation data, we aim to fix the temperature for the self-self attention so that it performs well without requiring hyperparameter tuning. Therefore, we propose an adaptive temperature using the average norm of the visual tokens before projection times the temperature originally used to train ViT: $\tau = \frac{N \cdot \sqrt{d}}{\sum_i ||x_i||_2}$. Further details on temperature ablation are available in Section 6.4.

**qkv-Ensemble:** We finally ensemble the iterative self-self attention applied to the query, key, and value projection to get a good performance trade-off between the different projections. The output $O_{qkv}$ of the proposed qkv-ensemble attention is formally described as follows:

$$O_{qkv} = \frac{(O_{qq} + O_{kk} + O_{vv})}{3} \qquad (6)$$

where $O_{qq}, O_{kk}, O_{vv}$ are the outputs based on the respective projection matrices $W_q, W_k, W_v$. Table 1 demonstrates the improvement achieved by applying normalization as well as ensembling over all possible projections.

## 4.2 SELF-SELF ATTENTION FOR CLUSTERING

Practically, self-self-attention calculates the similarity between each visual token and every other visual token. These similarities are then employed as weights in a weighted sum operation used to update the tokens. As a result, tokens are updated with a weighted sum of tokens, with more weight on more similar tokens, which may result in a respective mean representation corresponding to a cluster center. To validate this assumption, we conducted a simulation based on a set of 20 d-dimensional random Gaussian vectors representing the input token $x$ and a random linear projection

| Projection | Norm. | VOC | Context |
|---|---|---|---|
| CLIP | - | 10.4 | 7.7 |
| value | ✗ | 41.9 | 30.5 |
| key | ✗ | 43.9 | 31.0 |
| query | ✗ | 43.8 | 30.8 |
| qkv | ✗ | 43.1 | 30.7 |
| value | ✓ | 44.4 | 31.9 |
| key | ✓ | 44.8 | 32.0 |
| query | ✓ | 44.7 | 31.5 |
| qkv | ✓ | 45.1 | 32.3 |

Table 1: mIoU for v-v, k-k, and q-q attention and qkv ensemble on PascalVOC and PascalContext with and without $L^2$-Norm and adaptive temperature.

| Method | K | iter | VOC | Context |
|---|---|---|---|---|
| CLIP | - | - | 10.4 | 7.7 |
| Kmeans | 2 | - | 42.9 | 27.4 |
| | 3 | - | 44.1 | 30.0 |
| | 5 | - | 43.5 | 31.0 |
| | 7 | - | 43.4 | 31.1 |
| | 10 | - | 42.9 | 30.9 |
| GEM | - | 0 | 45.1 | 31.5 |
| GEM | - | 1 | 45.5 | 32.6 |
| GEM | - | 2 | 46.2 | 31.9 |
| GEM | - | 3 | 45.6 | 31.1 |

Table 2: Comparison of using Kmeans instead of the self-self-attention in the GEM architecture.



Figure 2: Comparison of self-self attention with Kmeans clustering on 20 vectors.

as $W_{proj}$. We iteratively apply the proposed self-self-attention on the 20 vectors, including normalization and with different temperature parameters. As shown in Figure 2, this process leads to a clustering of the 20 vectors using self-self attention. Moreover, it shows that high temperature, as well as more iterations, lead to few large clusters, while fewer iterations and a lower temperature enforce more and smaller clusters, thus controlling the granularity of the clustering. To further validate this assumption, we substitute the proposed self-self attention with an actual K-means clustering algorithm in the GEM architecture (for implementation details see Appendix A.1). Table 2 shows that Kmeans provides an acceptable performance but also that self-self attention outperforms Kmeans. We further analyze the impact of iteratively applying self-self attention. As shown in Figure 2, using more iterations leads to fewer distinct clusters; this improves performance on simpler datasets such as PascalVOC, while fewer iterations work better on more complex datasets such as PascalContext. To allow generalizability, we fix the number of iterations to 1 unless stated otherwise.

## 5 ANALYSIS OF LOCALIZATION PROPERTIES

In the following, we examine the factors contributing to the localization performance of the proposed method. We assume that for localization in VL models, two essential properties must be fulfilled: visual distinctiveness, which refers to the meaningful grouping of visual feature representations, and vision-language alignment, which entails the alignment of these groups with their respective textual descriptions encoded by the language model. In the case of CLIP, vision-language alignment translates to aligning patch tokens with the ViT [CLS] token, as the [CLS] token was trained to correlate with text embeddings through contrastive learning.

### 5.1 VISUAL DISTINCTIVENESS

To capture the visual distinctiveness, we consider two metrics.

**Patch-Patch Similarity.** This captures the similarity among patches within each layer. We define an overall path-patch similarity as $S_{pp} = \frac{1}{n(n-1)} \sum_{\substack{i,j \\ i \neq j}} x_i \cdot x_j^T$.

An increase in path-patch similarity indicates a higher tendency for tokens to share similar characteristics. However, high global path-patch similarity can indicate that all patch tokens are near-identical, thus reducing localization effectiveness.

**Object-Background Contrast.** We, therefore, further consider the object-background contrast. A critical characteristic of a model's localization proficiency is the ability to ensure similarity among patch tokens representing the same object while maintaining separation between those representing distinct objects. This characteristic permits the formation of semantically coherent clusters within the embedding space. To this end, we adapt the Michelson contrast to measure the contrast in the similarity between foreground and background patch tokens. For this evaluation, we leverage the segmentation masks of the training set of the PascalVOC dataset (Everingham et al., 2010). For a

given segmentation mask $M$ of an object, we first compute the overall inside-to-inside similarity (noted $S_{in,in}^M$) and inside-to-outside ($S_{in,out}^M$):

$$S_{in,in}^M = \frac{1}{m(m-1)} \sum_{\substack{i,j \in M \\ i \neq j}} \cos(x_i, x_j)^+, \quad S_{in,out}^M = \frac{1}{m(n-m)} \sum_{\substack{i \in M \\ k \notin \mathcal{M}}} \cos(x_i, x_k)^+ \quad (7)$$

Here, $m = |M|$ is the area covered by the mask, and the positive part function is employed to clamp negative similarities to zero, *i.e.* $\cdot^+ = \max(0, \cdot)$. The object-background contrast ($C^M$) for an object mask $M$ is then defined as:

$$C^M = \frac{S_{in,in}^M - S_{in,out}^M}{S_{in,in}^M + S_{in,out}^M} \quad (8)$$

We average across all the masks in the dataset: $MC^M = \frac{1}{|\mathcal{M}|} \sum_{M \in \mathcal{M}} CS^M$, with $|\mathcal{M}|$ being the total number of masks. Note that the ground truth masks are only used for analysis here.

## 5.2 VISION-LANGUAGE ALIGNMENT

Second, we consider the problem of vision-language alignment. Here, we aim to measure the contrast between the similarity of the text embedding representation of the class and the foreground patch embeddings, compared to the similarity of the text embedding and the background patches.

**Text-Object-Background contrast.** Let $p \in \mathbb{R}^{n \times d}$ be the patch token outputted by the vision transformer, where $n$ is the number of patches. For a segmentation mask $M$, the associated class name is denoted as $c(M)$, and we denote $t_{c(M)} \in \mathbb{R}^{1 \times d}$ the text embedding of that class. We compute the overall text-object similarity (noted $TS_{txt,obj}^M$) and text-background similarity ($S_{txt,bg}^M$):

$$TS_{txt,obj}^M = \frac{1}{m} \sum_{i \in M} \cos(t_{c(M)}, p_i)^+, \quad TS_{txt,bg}^M = \frac{1}{n-m} \sum_{k \notin \mathcal{M}} \cos(t_{c(M)}, p_k)^+ \quad (9)$$

The text-object-background contrast for mask $M$ is then defined as: $TC^M = \frac{TS_{txt,obj}^M - TS_{txt,bg}^M}{TS_{txt,obj}^M + TS_{txt,bg}^M}$

This metric is subsequently averaged across all masks in the dataset to derive the global text-object-background contrast $MTC = \frac{1}{|\mathcal{M}|} \sum_{M \in \mathcal{M}} TC^M$.

A higher positive value for $MTC$ signifies that foreground patch embeddings are closer to their corresponding text embeddings than background patch embeddings. A negative value would indicate an inverse relationship.

## 5.3 ANALYSIS

Figure 3 shows the results for the described metrics for CLIP, CLIPSurgery, and GEM for different numbers of iterations. The observed increase in patch-patch similarity from CLIP to CLIPSurgery, in figure 3a, is due to the clustering induced by the self-self attention. We contribute the slight decrease for GEM to the added normalization and temperature. This is recovered by the higher object-background contrast of GEM over CLIPSurgery and CLIP, pointing to the effective clustering of visual tokens and their ability to distinguish between distinct objects. Finally, the analysis of text-object similarity demonstrates improved alignment between visual tokens and text embeddings, enhancing vision-language integration. Notably, CLIP, while exhibiting similar levels of visual distinctiveness in terms of patch-patch similarity and object-background contrast, significantly lags in terms of vision-language alignment, showing a negative text-object contrast, which means that background patches tend to align more closely with object-class text embeddings. This aligns with earlier findings in Li et al. (2023b) and Mukhoti et al. (2023).

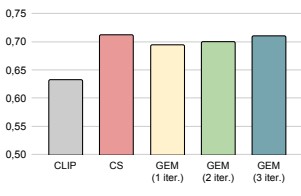 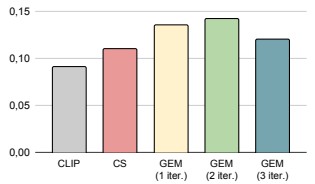 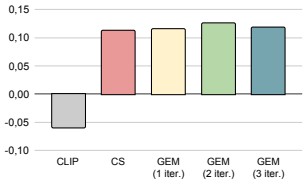

(a) Patch-patch similarity    (b) Object-Background Contrast    (c) Text-Object Contrast

Figure 3: Metrics to analyze the localization properties of CLIP, CLIPSurgery, and our method GEM. Each metric is computed on the training set of the PascalVOC dataset.

## 6 EVALUATION

### 6.1 SETUP

**Datasets.** The **PascalVOC** (Everingham et al., 2010) dataset provides segmentation masks for 20 classes in natural images, mainly focusing on common objects like cats, dogs, cars, and planes. A typical image contains 1.5 classes on average. Following previous works (Li et al., 2023b), (Zhou et al., 2022), we evaluate on the validation set. The **Pascal Context** (Mottaghi et al., 2014) dataset extends PascalVOC to 59 classes, supplemented by a background class. Compared to PascalVOC, it provides dense annotations for the whole scene. We use the test set comprising of $5,104$ images.The **OpenImages-V7.** (Benenson & Ferrari, 2022) dataset provides various annotations for a large set of images with a diverse spectrum of objects and real-world scenarios. For the following evaluation, we leverage the point-wise annotations of the validation set, with 36,702 images featuring 5,827 distinct class labels. For each image, a set of positive and negative point annotations is given. We only consider the positive point annotations here.

**Implementation.** For all experiments, we use the original pretrained weights as provided by the authors of the respective works, namely CLIP (Radford et al., 2021), OpenCLIP (Cherti et al., 2023), an open-source replication of CLIP, and BLIP (Li et al., 2022a).We apply the GEM architecture with the proposed adaptive temperature and one iteration for all datasets and models. We compute a dense semantic segmentation prediction for each image as follows: For each patch we compute the cosine similarity between the patch token of the vision encoder and the text embedding of the dataset class names. We use the prompt *'a photo of a {class name}'* to receive the text embedding. Finally, we upsample the predictions to the input image size via bilinear interpolation. When the input image is larger than the one used during the VL model training, we adapt the learned positional embeddings via bicubic interpolation. Note that ***we do not perform any retraining*** making our method essentially training-free.

**Evaluation.** Following common practices, we report the mean Intersection over Union (mIoU) for all datasets. For open-vocabulary semantic segmentation, we follow (Xu et al., 2022) and resize each input image to have a shorter side length of 448. For PascalVOC we predict only the foreground classes and get the background by thresholding the softmax-normalized-similarity between the patch tokens and the text embedding of each class using a fixed threshold of 0.8. For Pascal Context, we follow common practices and evaluate only on the 59 foreground classes. For OpenImages-V7, for each positive class in the image, we min-max normalize the prediction and use a fixed threshold of 0.5 to obtain the predicted mask. We follow the authors' guidelines (Benenson & Ferrari, 2022) and compute the IoU over the sets of positive and negative ground-truth points for all positive classes in the respective image.

### 6.2 COMPARISON TO STATE-OF-THE-ART

We evaluate our approach against three groups of state-of-the-art methods in open-vocabulary segmentation. First, we compare with methods that perform training-free zero-shot segmentation, namely MaskCLIP, MaskCLIP[2], and CLIPSurgery.

| method | encoder | model | dataset pretraining | annotation | train free | VOC | mIoU Context | V7 |
|---|---|---|---|---|---|---|---|---|
| SPNet (Xian et al., 2019) | ResNet101 | scratch | COCO, VOC, Context | SM | ✗ | 15.6† | 4.0† | - |
| ZS3Net (Bucher et al., 2019) | ResNet-101 | scratch | VOC, Context | SM | ✗ | 17.7† | 7.7† | - |
| OpenSeg (Ghiasi et al., 2022) | ENet-B7+FPN | ALIGN | COCO, Loc. Narr | IT, UM | ✗ | 72.2 | 48.2 | - |
| MaskCLIP[3] (Ding et al., 2022) | ViT-B/16 | CLIP | COCO | SM | ✗ | - | 45.9 | - |
| OVSeg (Liang et al., 2023) | ViT-B/16 | CLIP | COCO-Stuff-171 | UM | ✗ | 94.5 | 55.7 | - |
| CLIP-ES (Lin et al., 2023) | ResNet101 | CLIP | COCO-Stuff-171 | IC | ✗ | 75.0 | - | - |
| ViL-Seg (Liu et al., 2022) | ViT-B/16 | scratch | GCC | IT | ✗ | 34.4† | 16.3† | - |
| GroupViT* (Xu et al., 2022) | ViT-S/16 | scratch | GCC+YFCC | IT | ✗ | 42.8 | 15.1 | - |
| GroupViT (Xu et al., 2022) | ViT-S/16 | scratch | GCC+YFCC | IT | ✗ | 52.3 | 22.4 | - |
| OVSegmentor (Xu et al., 2023) | ViT-B/16 | DINO | GCC | IT | ✗ | 53.8 | 20.4 | - |
| SegCLIP (Luo et al., 2023) | ViT-B/16 | CLIP | CC, COCOcap | IT, IC | ✗ | 52.6 | 24.7 | - |
| PACL (Mukhoti et al., 2023) | ViT-B/16 | CLIP | WIT-400M +CC12M, YFCC | IT | ✗ | 72.3 | 50.1 | - |
| MaskCLIP[2] (Dong et al., 2023) | ViT-B/16 | scratch | YFCC | IT | ✓ | - | 17.2 | - |
| CLIP (Radford et al., 2021) | ViT-B/16 | CLIP | WIT-400M | IT | ✓ | 10.4 | 7.7 | 27.6 |
| MaskCLIP (Zhou et al., 2022) | ViT-B/16 | CLIP | WIT-400M | IT | ✓ | 28.6 | 23.8 | 42.0 |
| CLIP Surgery (Li et al., 2023b) | ViT-B/16 | CLIP | WIT-400M | IT | ✓ | 41.2 | 30.5 | 47.8 |
| GEM (our) | ViT-B/16 | CLIP | WIT-400M | IT | ✓ | 46.2 | 32.6 | 50.9 |

Table 3: **Comparison to state-of-the-art methods**: Models marked with † are evaluated under relaxed constraints, specifically on a subset of unseen classes. * signify our evaluation. We use the following short form, GCC: Google Conceptual Captions 12M, YFCC: YFCC15M, COCO: COCO2017, RCOCO: RefCOCO, RCOCO+: RefCOCO+, CC: Conceptual Captions, SBU: SBU Captions, VG: Visual Genome, COCOCap: COCO Captions, VOC: PascalVOC, PCont: PascalContext and V7: OpenImagesV7. SM: segmentation mask, IT: image-text, IC: image caption, UM: unlabeled mask, IC: image classes.

Second, we report the performance of models trained explicitly for segmentation on image-caption pair annotations, i.e., GroupViT (Xu et al., 2022), OVSegmentor (Xu et al., 2023), SegCLIP Luo et al. (2023), and ViL-Seg (Liu et al., 2022). Additionally, we consider PACL (Mukhoti et al., 2023) in this group, which trains a decoder on top of CLIP using a loss designed for patch grouping. Third, as an upper bound, we provide results for methods trained with some form of labeling, e.g. segmentation masks, such as OpenSeg (Ghiasi et al., 2022), CLIP-RIS (Yu et al., 2023), MaskCLIP[3] (Ding et al., 2022), and OVSeg (Liang et al., 2023). We report the mIoU In Table 3. It shows that the proposed method consistently outperforms all training-free approaches. It further exhibits competitive performance compared to models tailored specifically for localization on the more complex dataset PascalContext surpassing all other models except PACL. In Figure 4, we present qualitative results showcasing the efficacy of our methods (see Appendix A.4 for more qualitative examples).

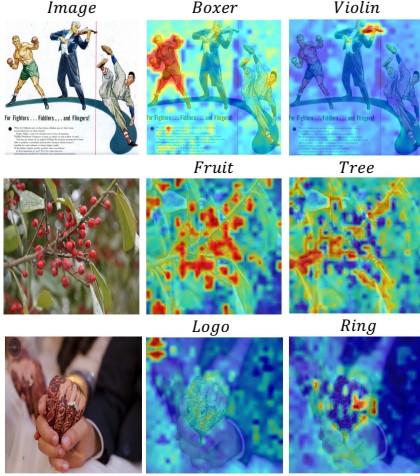

Figure 4: Qualitative examples from the OpenImagesV7 dataset of GEM on ViT-B/16 on the CLIP backbone. Additional examples can be found in the appendix A.4.

## 6.3 TEMPERATURE

Finally, we regard the impact of the adaptive temperature. Figure 5 provides an overview of the segmentation performance of GEM-CLIP under varying temperature settings for self-self attention for two different ViT sizes, namely ViT-B/16 and ViT-B/32, and two datasets, PascalVOC and PascalContext. Specifically, we evaluate the performance using the proposed "adaptive temperature" multiplied by a variable factor. We observe that, across different models and datasets, the highest mIoU is consistently achieved when the varying temperature is equal to 1, indicating the effectiveness of our proposed heuristic outlined in Section 4, i.e., $\tau = \frac{N \cdot \sqrt{d}}{\sum_i ||x_i||_2}$. This result underscores the robustness and generalizability across various models, as it adapts to the specific characteristics of the input vector, dependent on the model employed.

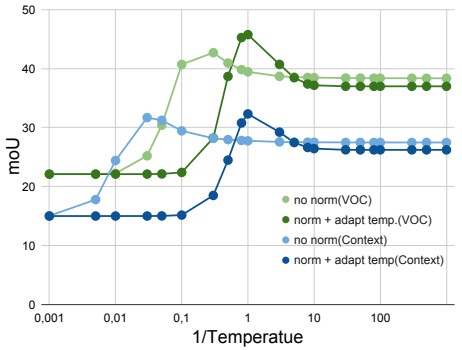 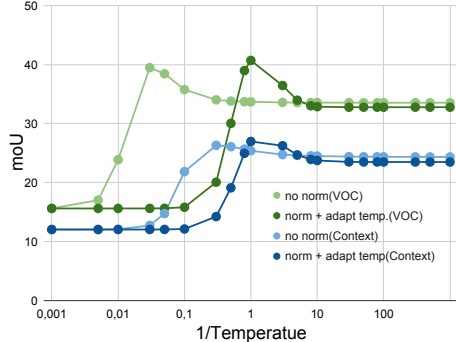

Figure 5: Evaluation of localization performance for CLIP ViT-B/16 (left) and ViT-B/32 (right) for the PascalVOC and PascalContext dataset with normalization and adaptive temperature as well as without. It shows that the results peak at the proposed temperature (1).

## 6.4 ARCHITECTURE AND MODEL SIZE

We further extend our analysis beyond the ViT-B/16 model to explore the generalizability of our findings across various sizes of CLIP, including ViT-B/32 and ViT-L/14, as well as to other VL backbones, specifically OpenCLIP, BLIP, and CLIPA. OpenCLIP, as an open-source replication of CLIP, thus to investigate the generality on an architecture closed to CLIP, while BLIP is trained with a multi-task objective, and, different from CLIP and OpenCLIP, encompassing image-text matching, image captioning, and image-text contrast. Table 4 shows the results for the VL models and different backbones. It shows that the GEM method consistently improves localization performance across all model sizes. As expected, for a fixed ViT-B size, increasing the patch size from 16 to 32 reduces the performance slightly. We further observe that larger ViT-L encoders do not yield better localization performance. Specifically, GEM-ViT-B/16 consistently outperforms its larger counterparts GEM-ViT-L/14. Finally, BLIP, as the only model trained with multi-objectives, tends to perform less in localization than models trained solely with an image-text contrastive loss.

| Backbone | Model | GEM | VOC | Context | V7 |
|---|---|---|---|---|---|
| ViT-B/16 | CLIP | ✗ | 10.4 | 7.7 | 27.6 |
| | OpenCLIP | ✗ | 13.6 | 7.5 | 26.3 |
| | BLIP | ✗ | 4.4 | 3.1 | 29.4 |
| | CLIP | ✓ | 46.2 | 32.6 | 50.9 |
| | OpenCLIP | ✓ | 43.1 | 31.7 | 49.9 |
| | BLIP | ✓ | 42.8 | 23.5 | 45.2 |
| ViT-B/32 | CLIP | ✗ | 4.8 | 3.3 | 28.1 |
| | OpenCLIP | ✗ | 9.5 | 5.4 | 27.3 |
| | CLIP | ✓ | 40.5 | 27.0 | 46.6 |
| | OpenCLIP | ✓ | 39.3 | 23.9 | 45.5 |
| ViT-L/14 | CLIP | ✗ | 4.1 | 3.8 | 29.1 |
| | OpenCLIP | ✗ | 6.6 | 2.9 | 27.0 |
| | BLIP | ✗ | 7.2 | 3.6 | 29.7 |
| | CLIP | ✓ | 44.6 | 28.6 | 46.3 |
| | OpenCLIP | ✓ | 40.0 | 27.5 | 42.4 |
| | BLIP | ✓ | 32.1 | 21.4 | 44.9 |

Table 4: Evaluation of the GEM architecture on various pretrained VL backbones. It shows that the method improves across all backbones and architectures with better performance for backbones trained image-text contrastive loss only (CLIP, OpenCLIP) as well as for smaller patch size (ViT-B/16 compared to Vit-B/32) and architecture (ViT-B compared to ViT-L).

## 7 DISCUSSION

In this work, we introduce the Grounding Everything Model, leveraging the latent localization capabilities of VL models trained on web-scale datasets. We propose a self-self attention pipeline for extracting localization information from VL models, complemented by a set of regularizations to ensure generalizability across diverse VL models and datasets. GEM effectively enables open-vocabulary localization without the need for additional training.

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

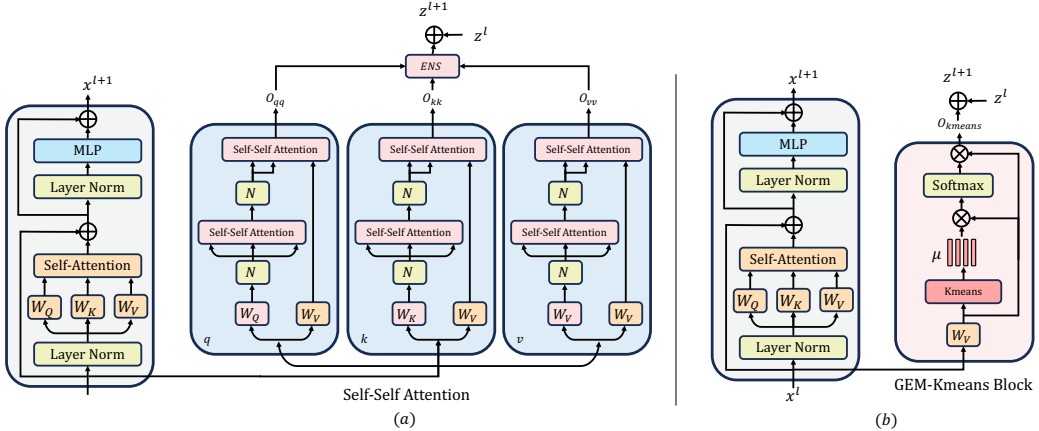

Figure 6: *(a):* Detailed Illustration of GEM for a number of iterations for the iterative self-self attention equal to 1, where the block $N$ corresponds to $L^2$ normalization. *(b):* Detailed Illustration of GEM-Kmeans block. The self-self attention is replaced by a Kmeans directly applied over the values.

# A    APPENDIX

In this section, we provide additional implementation details, analysis, ablation and qualtitatives examples. In Appendix A.1, detail the implementation of the proposed GEM, as well as the implementation of the variant where we replace the self-self attention by a Kmeans and give additional infromation on the clustering in Appendix A.2. Then, in Appendix A.3 we provide further ablation analysis. Finally, in Appendix A.4 we perform a qualitative comparison of our model to CLIP, CLIPSurgery and MaskCLIP.

## A.1    ADDITIONAL IMPLEMENTATION DETAILS

**Architecture and Hyperparameters:** GEM is built in parallel to the vision transformer by processing input features coming from the vision transformer through a series of ensembled iterative-temperature regularized self-self attention. For reproducibility, we fix the number of iterations of self-self attention to 1, i.e., that we do one step of self-self attention applied to the projected features **and** one step of self-self attention applied to the values. We also used the temperature heuristic proposed section for all experiments. Figure 6.a, provides a detailed overview of the proposed method.

**GEM-Kmeans:** In section 4.2 we consider the connection between the proposed self-self attention and clustering. To substantiate that connection we replaced the self-self attention with a Kmeans algorithm. Figure 6.b provides implementation details. Given input visual tokens $x \in \mathbb{R}^{n \times d}$ and the value projection matrix $W_v \in \mathbb{R}^{d \times d}$ (from the pretrained ViT), the value tokens are computed as $v_i = x_i W_V$ Kmeans output a set of $K$ centroids denoted $\boldsymbol{\mu} = (\mu_1, \mu_2, \ldots \mu_K) \in \mathbb{R}^{K \times d}$. The output $O_{kmeans}$ is then obtained by replacing each value token with a weighted sum of the computed centroids:

$$Attn_{kmeans} = \mathrm{softmax}(v_i \cdot \boldsymbol{\mu}^T), \quad O_{kmeans} = Attn_{kmeans} \cdot \boldsymbol{\mu}^T \qquad (10)$$

Note the similarity of the above equation with equation 5 that describes the output of our self-self attention.

## A.2    FURTHER DETAILS ON CLUSTER ANALYSIS

In section 4.2, we evoque the idea that our self-self attention seems to act as a form of clustering. In Figure 7 we extend the simulation presented in section 4.2 to more iterations and temperatures. We can observe that when we increase the number of iterations (from top to bottom), the cluster memberships become crisper and fewer clusters tend to form.

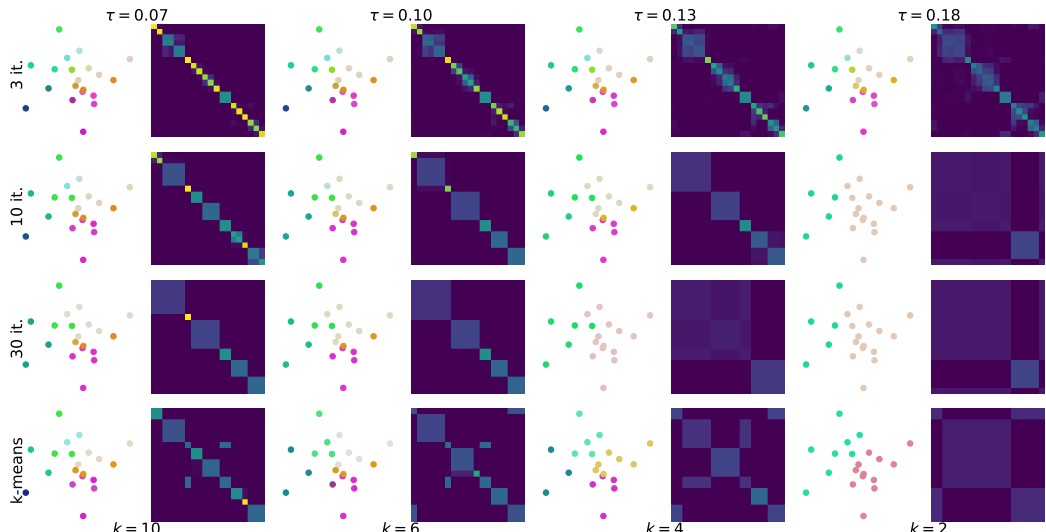

Figure 7: Comparison of self-self attention with traditional Kmeans clustering on a set of 20 vectors. Utilizing cosine distance and varying parameters, our results demonstrate that as the number of iterations increases, self-self attention produces crisper cluster. Self-Self Attention-based Clustering: In the top 3 rows, a set of 20 vectors is softly clustered using self-self attention clustering. In the bottom row, k-means is used for clustering. Clustering is performed based on cosine distance. From left to right, we increase $\tau$ or reduce the number of clusters $k$, respectively. Displayed are the 20 data points (reduced to two dimensions via PCA) and their color represents a smooth cluster membership (the vector into which they are transformed is translated into a color value.) Further, the attention matrix is displayed for each clustering (the points were manually ordered for visual simplicity.) We can observe that when we increase the number of iterations (from top to bottom), the cluster memberships become crisper and fewer clusters tend to form.

## A.3 ADDITIONAL ABLATION

To gain a deeper understanding of the factors influencing the performance of our method, we conducted additional ablation. Namely, we disentangle GEM's performance for the depth of the vision transformer at which we apply self-self-attention. We also evaluate the effect of adding the MLPs from the vision transformer encoder after the self-self attention in the alternative pathway.

**Impact of path length:** In Table 5 we evaluate the segmentation performance of GEM applied to CLIP for two model sizes (ViT-B/16 and ViT-B/32) for different starting depths for the alternative pathway. For ViT-B/16, the performance remains sensibly the same as long as GEM is applied after the last layer. This can be explained by the fact that the skip connection of GEM's alternative pathway is essentially an exponential moving average of GEM applied at each layer. Therefore, the influence on the output features of the first layers decays exponentially. Therefore, we fixed the depth $d$ of GEM to equal to $d = 4$.

**Impact of MLP:** Originally, the studied VL models were trained using MLPs in their transformer blocks. Contrarily, in GEM's pathway the MLPs are not used, and therefore we need to assess the influence of these MLPs on the downstream performance. Table 5 shows the influence of adding the MLP from the pretrained VL in our method. We can see that MLPs have a slight negative effect on the downstream performance. In the case of CLIP, we attribute that to the fact that to compute the final loss only the ViT's [CLS] token is used, and the patch tokens are discarded. Hence, the last MLP had a supervision signal steaming from only the [CLS] token, hence, this module was never trained to process patch tokens.

## A.4 QUALITATIVE ANALYSIS

**Failure Cases:** Figure 8 shows some failure cases of GEM. For the first image, when prompted with the text description *"Humain body"*, the model segments both the human body and the vehicle body. For the second image, prompted with *"Vehicle registration plate"*, the model focuses on the whole car instead of only localizing the registration plate. It seems that the text encoder is giving more

| Backbone | MLP | 1 | 2 | 3 | 4 | 5 | 6 | 7 | 8 | 9 | 10 | 11 |
|----------|-----|-----|------|------|------|------|------|------|------|------|------|------|
| ViT-B/16 | ✗ | 4.8 | 43.8 | 45.2 | 45.5 | 45.6 | 45.3 | 45.5 | 45.5 | 45.4 | 45.4 | 45.1 |
| ViT-B/16 | ✓ | 26.2 | 38.8 | 42.4 | 42.1 | 42.2 | 42.3 | 41.9 | 41.6 | 41.7 | 42.0 | 41.6 |
| ViT-B/32 | ✗ | 5.1 | 26.1 | 40.3 | 41.5 | 41.4 | 41.3 | 41.2 | 41.2 | 41.1 | 41.0 | 41.0 |
| ViT-B/32 | ✓ | 4.3 | 21.6 | 38.4 | 40.3 | 40.2 | 40.1 | 40.1 | 40.0 | 39.7 | 39.6 | 39.7 |

Table 5: Impact of depth.

weight to certain words, to "body" in the first example and to "vehicle" in the second. This way, when compared to the visual features the model returns the localization of the "emphasized word". This effect can be mitigated by removing or replacing the "emphasized" word with another more descriptive, as shown Figure 8. Therefore, we attribute this type of failure case to the text encoder, paving the way for future research.

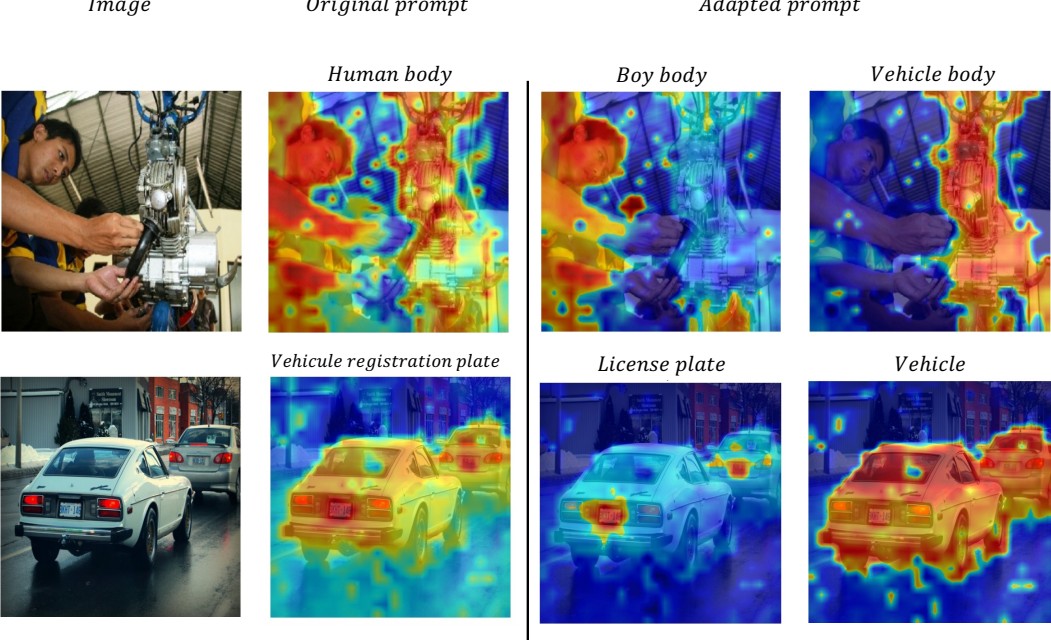

Figure 8: Failure cases of GEM from the OpenImagesV7 dataset.

**Qualitative examples:** Figure 9 presents qualitative examples for CLIP, CLIPSurgery, MaskCLIP and our method GEM. In most cases, our method is able to localize the prompted object accurately. The figure also points out the enhanced localization ability of GEM over CLIPSurgery, notably row 2, where CLIPSurgery fails to localize the class *"man"*. Overall, GEM produces more precise localization than CLIPSurgery and MaskCLIP. Interestingly, while vanilla CLIP excels at identifying which classes are present (in a zero-shot classification setting Radford et al. (2021)) in images, it falls short in localizing them effectively, underscoring the advantages of our method.

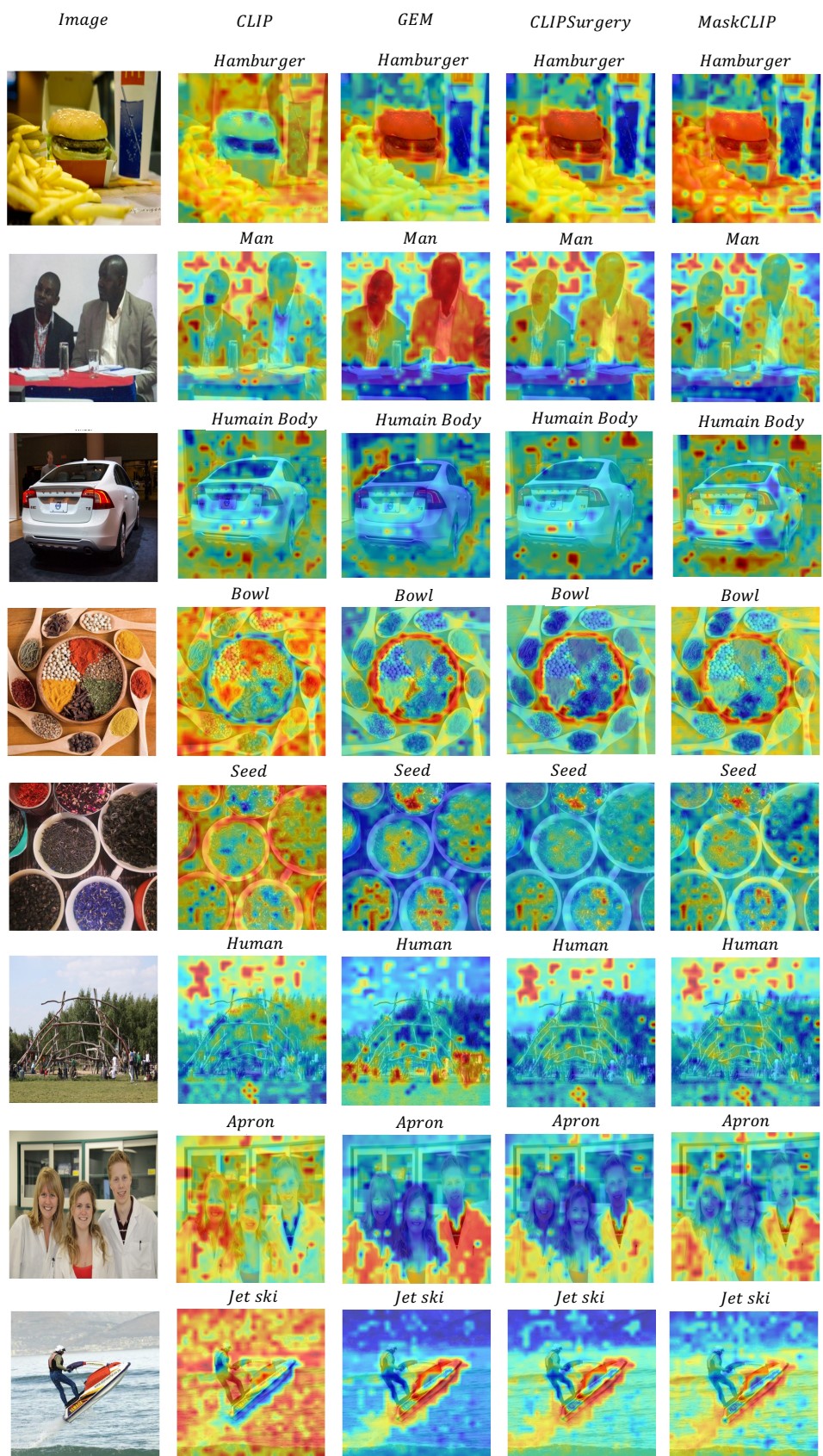

Figure 9: Qualitative examples for CLIP, GEM, CLIPSurgery, and MaskCLIP from the OpenImagesV7 dataset. All methods use ViT-B/16 as backbone.

