# OpenReview forum: "Grounding Everything: Emerging Localization Properties in Vision-Language Transformers"
_ICLR.cc/2024/Conference — ICLR 2024 Conference Withdrawn Submission_

### Official Review · Reviewer_Ursz · 2023-10-31

**Soundness:** 3 good
**Presentation:** 3 good
**Contribution:** 1 poor
**Rating:** 1
**Confidence:** 4

**Summary:**

The paper proposes a post-hoc (no training) way to cluster features, and shows some applications to open-vocabulary segmentation.
The paper takes the value-value attention in CLIPSurgery and generalizes it to iterated value-value, key-key, query-query attention (or a combination thereof) to compute similarity.
Experiments are on object localization, PascalVOC, PascalContext, and OpenImages-V7, where the method beats raw similarity between text and features. There is a nice comparison to k-means clustering, where results are a bit better than K-means (but quite similar nonetheless).

**Strengths:**

### Presentation
The paper is well-written.

### Experiments
I think the experiments are well done (though some comparisons are missing, see weaknesses). In particular, I like the insightful comparison to K-means.

Experimental results evaluated on multiple architectures and datasets. There are clear ablations of the different components and settings (temperature, iterations, self-self attention).

### Method
The method is simple, and reasonably principled

**Weaknesses:**

Overall I'm not seeing a use case for this method. K-means is a quicker-to-implement clustering/visualization method, and on the other end of the spectrum other techniques like DeepCut seem to offer better performance.

### Presentation
The title is "Grounding Everything Model". I think at best this is not a good fit for the paper, and at worst has the possibility to be a bit misleading. Firstly, this isn't a model, right? It is a post-hoc clustering technique and the authors evaluate it on several pretrained models. I suppose this is a form of grounding because it is a way to localize queries in an image, when the original architecture doesn't support that. But if you want to ground something, there are better models -- e.g. [Grouding-Dino](https://arxiv.org/abs/2303.05499)

### Experiments
1. The paper misses a lot of existing literature -- what about comparison to other papers that do clustering of pretrained features (e.g. [Deep Spectral Methods](https://github.com/lukemelas/deep-spectral-segmentation)? The only other clustering method compared to is K-means (and I guess CLIPSurgery, which I believe is just 1 iteration of V-V attention).
2. The results are pretty weak -- not much better than K-means. Other feature-clustering approaches seem to have much better results -- e.g. [DeepCut](https://sampl-weizmann.github.io/DeepCut/). That paper also has extensive comparisons to existing lit that clusters deep features on VOC and other datasets.

**Questions:**

1. I wonder if this could be understood as an iterative approximation to some type of spectral clustering (computing neighbor laplacian, computing eigenvectors, the temperature setting in the alg washes out all eigenvectors with eigenvalues < some thresh, and then run k-means clustering on the eigenvectors). An exact analysis is escaping me here.
2. For the random gaussian clustering technique -- all vectors come from the same gaussian, right? And the resulting clustering from GEM is similar to that of K-means. That is a somewhat arbitrary clustering because the underlying process all comes from the same cluster. Why not generate data from different clusters, and show that GEM is better able to recover the underlying clusters?

---

> ### Author Response · Authors · 2023-11-13
> **Response to Ursz**
>
> Thanks so much for the review, highlighting the good points regarding presentation, experiments, and methods, but also offering clear criticism. Please give us a chance to clarify the raised concerns in the following:
>
> ### Weaknesses
>
> ### Presentation
> *Title is not a good fit - it's not a model:* We would politely challenge this assumption and argue that the paper indeed proposes an architecture (at least a new layer design) that allows to leverage pretrained weights for localization. As there is a general understanding that a model is a set of pretrained parameters that together with a predefined architecture produce a certain output, we would argue that in our case, while we don't change the parameters, we change the architecture, thus producing a different output compared to the regular transformer. We are also open to changing the name of GEM to "Grounding Everything Module" rather than "Grounding Everything Model".
>
> ### Experiments
> 1. Difference to Grounding-Dino and Deep Spectral Methods:
> Grounding Dino and Deep Spectral Methods are both methods that rely on self-supervised trained image backbones. As a result, the image backbone itself does not provide any image-text alignment. This has to be established in a post-processing step:
> (1) In case of Deep Spectral Methods by Hungarian matching between the detected segments and the GT segments ("For semantic segmentation (2b), we convert the eigenvectors into discrete segments, compute a feature vector for each segment, and cluster these segments across an entire dataset."). The method as-is can thus not be used for free-text prompting.
> (2) In case of Grounding Dino, the alignment between image and text needs to be established in a separate training cycle making it not training-free. Further, this training is so far only done for bounding box based detection. As a result, there are currently no semantic segmentations baselines available for Grouding Dino that would allow a direct comparison. Thus, the claim that "if you want to ground something, there are better models -- e.g. Grouding-Dino" can currently not be validated. (Please see also our answer to reviewer uWg6 (Motivation) on Grounding SAM)
>
> Finally, we want to highlight that the proposed method is dealing with exactly this problem, how can we process visual tokens through the architecture while preserving a tokenwise alignment between the single tokens and the text representation. Neither Grounding Dino nor Deep Spectral Methods are considering this problem.
>
> 2. The results are pretty weak -- not much better than K-means.
> While we agree that the K-means approach performs almost comparable to the proposed approach, we have to point out that here, not our approach has been modified to achieve the best results (best results at 1-2 iterations (45.5/46.2 for VOC and 32.6/31.9 for Context), but it is actually the K-means approach that needs an appropriate hyper-parameter tuning per dataset to perform comparable (best results at 3 iters for VOC (44.1), 7 iter for Context (31.1)). Note that again, to tune this for each dataset, we would again need GT annotation, while GEM with 2 iterations provides good (not-optimized) results across all tested dataset.
>
> ### Overall
> K-means is a quicker-to-implement: While this assessment might depend on individual programmer skills (and should not to a major criterion for a new method), we want to point out that beyond the disadvantages mentioned earlier, K-means has a significantly higher runtime (depending on the number of clusters used for Kmeans). Kmeans is not only slower but also requires to choose the optimal K.
>
> Please let us know if you have any other questions or concerns regarding our paper or response.
> If we have successfully addressed your questions, we would highly appreciate an increased score.

---

### Official Review · Reviewer_PF2u · 2023-10-31

**Soundness:** 2 fair
**Presentation:** 2 fair
**Contribution:** 1 poor
**Rating:** 5
**Confidence:** 3

**Summary:**

This work proposes a training-free method to address the image referential localization problem. It proposes GEM model and extends the v-v attention to self-self attention. The experimental results show that the proposed GEM framework improves the performance in training-free open-vocabulary localization.

**Strengths:**

1. This paper proposes a training-free method to address the open-vocabulary segmentation problem, and it improves the baseline model.
2. This paper shows that by merely computing the proposed heuristic similarity and without training, the pre-trained model already does well in open-vocabulary segmentation task.

**Weaknesses:**

1. This work claims the localization property is 'emerging', which is somehow overclaimed. The resulting model yields merely less than 50% mIoU in VOC, which is far from 'emerging' performance.
2. The so-called localization is usually used for a high-level concept. In this work, it should be replaced by the more precise term 'segmentation'.
3. The overall method is straightforward and tedious. It shows a way to compute a heuristic similarity with a high distinctiveness and does improve the performance. However, the procedures are too tedious and there is still a huge gap with the methods involved with training.
4. There is always a consensus that the final model usually needs two phases --- pre-training and finetuning. Why do we need a finetuning-free method for a segmentation task? A model pre-trained with a large quantity of data needs finetuning or alignment for a specific task unless it already has the emergent ability for multiple tasks. But today, we have seen that there is still a huge performance between a pre-trained model and a finetuned one, even in a single task. In this way, the finetuning-free property does not really matter in the context of a single task. The authors should elaborate more on this.

**Questions:**

The authors should give their explanations for each weakness above.

Overall, I think this research is not really beneficial and promising for the development of this area, and this is not a paper that reaches the acceptance standard of ICLR.

---

> ### Author Response · Authors · 2023-11-13
> **Response to PF2u**
>
> Thank you for taking the time to review the paper. We will try to address the critical points in the following:
>
> ### Weaknesses
> (1) *Overclaim on 'Emerging':* Thanks for this comment. [1] defines the term 'emergent' in context of foundation models as 'Emergence means that the behavior of a system is implicitly induced rather than explicitly constructed;'. For the proposed method, we decided to use this term as the here showed localization is not explicitly enforced at training time as the training is solely limited to contrasting global vision and language representations. We offer to include this reasoning in the introduction of the paper. If reviewer PF2u completely disagrees with this interpretation and the area chair supports agrees, we would offer to change the title to 'Grounding Everything: Leveraging Localization Properties in Vision-Language Transformers'.
>
> (2) *Localization vs segmentation:*
> We agree that there is a fluent transition between localization and segmentation here. In general, we would argue that the proposed method technically just computes correlations between visual tokens and a global text token. The output can therefore be understood as a scoremap or heatmap. In our understanding, this can be turned into a segmentation map by an additional post-processing step e.g. by thresholding or softmax, but our method e.g., never actually considers standard segmentation information like outlines, texture, etc and is also never trained to actually output segmentation maps. We would therefore argue that this is conceptually different enough to justify the term of localization for the general idea, and use this for the downstream task of segmentation in one case, and e.g. the problem of pointwise classification in another case as done for OpenImages v7. (see also our answer to uWg6 on Motivation)
>
> (3) *Too tedious:*
> We would politely challenge this assumption based on the following reasoning: First,  GEM has a runtime of about 0.028s per image on OV7 (VIT-B/16).  We agree that the second pathway creates additional overhead, but it is usually not necessary for the full transformer, but rather starts in the middle layers (see Tab 5 Appendix), or even later if a drop in performance of 1-2 percent is acceptable. We will provide a detailed runtime comparison with other methods in the final version of the paper.
>
> (4) *Huge gap with the methods involved with training ---  Why do we need a finetuning-free method for a segmentation task? :*
> We agree that in case of simple datasets with small vocabulary (VOC 20 classes, Context 59 classes), pretrained methods do better. But training usually requires annotation, which can only be provided for so many classes (which also have to be defined in advance to avoid too much annotation disagreement). A training-free method is not limited by this, and therefore outperforms those methods on large-scale real-world tasks such as OpenImage v7 (point-level annotations on 5827 classes). We report the respective numbers for GroupViT and SegCLIP that do not use mask annotation during training but are not training free and OVSeg that uses mask information during training:
>
> | Method   | OpenImagesV7 mIoU |
> | -------- | ------- |
> | GroupViT | 36.5    |
> | SegCLIP  | 32.1    |
> | OVSeg    | 22.5    |
> | -------- | ------- |
> | GEM      | 50.9    |
>
> Please let us know if you have any other questions or concerns regarding our paper or response.

---

### Official Review · Reviewer_67Tz · 2023-11-01

**Soundness:** 3 good
**Presentation:** 3 good
**Contribution:** 2 fair
**Rating:** 6
**Confidence:** 3

**Summary:**

The paper present Grounding Everything Model (GEM) in order use pre-trained Vision-Language (VL) models for object Localization without the need of re-training or fine-tuning the VL model. Building on CLIPSurgery (that incorporates value-value attention), the paper extends it to a generalized self-self attention path and propose a set of regularizations that allows the model to better generalize across datasets and backbones. Experimental results on three benchmarks (PascalVOC, Pascal Context, and OpenImages-V7) shows promising performance boost compared to existing training-free works.

**Strengths:**

The paper is well-written and easy to follow. Related literature is being reviewed in a comprehensive manner. The paper provides an in-depth analysis of the properties of self-self attention, along with its connection with k-means clustering, in various pretrained ViT transformer models.

**Weaknesses:**

My main point of criticism concerns the technical novelty of this work, since it builds on the recently proposed CLIPSurgery and proceeds by simply swapping value-value with self-self attention.

Whilst this might appear as a marginal technical contribution, the authors provides good experimental results and a thorough analysis of v-v and self-self attention, with an interesting connection to k-means. It is not totally convincing, though, how this work is essentially different than CLIPSurgery. I would expect a more comprehensive comparison to CLIPSurgery, especially in methodological terms -- why the proposed method's perfomance cannot be achieved by CLIPSurgery by simply exploring the attention mechanism?

**Questions:**

In Table 2, k-means achieves the best results for k=3 (PascalVOC) and k=7 (Pascal Context):
 - What would be a possible explanation for this?
 - How would a different clustering method affect the results (in other words., could a different clustering method achieve results than the proposed GEM's ones)?

---

> ### Author Response · Authors · 2023-11-13
> **Response to 67Tz**
>
> Thank you for reviewing our paper and highlighting the strengths. We try to clarify the difference to CLIPSurgery and the connection to K-means in the following.
>
> ## Weaknesses
> *I would expect a more comprehensive comparison to CLIPSurgery, especially in methodological terms -- Why the proposed method's performance cannot be achieved by CLIPSurgery by simply exploring the attention mechanism?*
> We would argue main things to do to achieve similar performance with CLIPSurgery, so value-value attention alone, are exactly the extensions proposed in the paper, mainly introducing a normalization and temperature regularization that allows to guide the self-self attention based clustering (see Tab. 1 results based on value-value attention). It further shows that ensembling also improves, but here, the question would be how to augment v-v attention alone with other valuable cues. As we showed q and k have similar or even better expressive power, those might be the most natural options in this case, which finally leads to the proposed pipeline.
>
> *Connection to K-means*
> Thanks for highlighting that. We think that (1) extending v-v to a generalized self-self attention formulation and (2) establishing the connection between self-self attention and clustering in general might be one of the valuable contributions of this work, as it provides a theoretical underpinning of the ideas  presented in CLIPSurgery and therefore might help to raise a better awareness for this line of research and hopefully pave the way for even better ideas and methods in this direction.
>
>
> ### Questions:
> *In Table 2, k-means achieves the best results for k=3 (PascalVOC) and k=7 (Pascal Context):*
>
> *What would be a possible explanation for this?*
> Thanks for raising this point. The accuracy in the case of clustering depends on the number of clusters as each dataset has objects labeled at different granularity, e.g., Pascal VOC has 1.5 objects labeled per image. If we assume that the background has to form another group, it is reasonable to assume that 3 clusters perform best. Context has on average 4.8 classes per image, and 5-7 clusters perform best here.
> Note that this is a hyperparameter that would need to be tuned for each dataset (or image) separately, thus actually needing some pre-knowledge or even annotation, which we can avoid with the proposed method. (Please also see the answer to Reviewer uWg6 on GEM vs. CLIPSurgery(CS))
>
> *How would a different clustering method affect the results (in other words., could a different clustering method achieve results than the proposed GEM's ones)?*
>
> We propose to include a further evaluation with more clustering algorithms in the camera-ready version. Overall, we would argue that most clustering methods depend on a set of hyperparameters, so we expect to have to tune each method separately, leading to the same disadvantages as k-means. If you have one or more specific method(s) in mind, please let us know.
>
>
> Please let us know if you have any other questions or concerns regarding our paper or response.

---

### Official Review · Reviewer_uWg6 · 2023-11-02

**Soundness:** 2 fair
**Presentation:** 3 good
**Contribution:** 2 fair
**Rating:** 5
**Confidence:** 4

**Summary:**

This paper introduces the Grounding Everything Model (GEM), leveraging the latent localization capabilities of VL models trained on web-scale datasets. In this paper, the authors present an extended version of the CLIPSurgery concept of v-v attention, called self-self attention, which can extract localization information from VL models. The GEM method effectively enables open vocabulary localization without additional training.

**Strengths:**

1) Overall, this paper is well-written, and the technical details are easy to follow.

2) The main idea of avoiding additional training for open vocabulary localization is unquestionably important.

3) The main contribution of this paper is the self-self architecture, which is an extension of CLIPSurgery's concept of v-v attention.

**Weaknesses:**

**Technical Novelty.** The fact that v-v attention was suggested without considering that it can also be applied to keys and queries seems odd. I believe the CLIPSurgery paper may have been lacking in ablation since a properly executed ablation should reveal that v-v attention is not necessarily the most effective approach. According to this paper, they "...discovered that the major cause of this issue is the parameters (query and key) in the self-attention module" (see CLIP Surgery). Therefore, it seems strange that the query and key parameters are actually required now.

This brings me to the main concern I have. The proposed architecture is a slight extension of the v-v attention, and in particular, I cannot see how duplicate self-attention, in a similar manner to v-v attention, could be applied to other tasks or models. According to my understanding, this represents a very small ablation that the first paper missed, and is not something the community could make significant use of in the near future for another task or dataset.


**Motivation.** It is understandable that the authors are motivated to use pretrained vision and language models for localization. However, I am unsure why this is important since we already have SAM [1] (and other improved versions), which demonstrate that these supervised models are capable of almost completely solving this problem. The fact that SAM performs extremely well on this task raises the question of whether pretrained vision and language are still required, given its excellent performance.

[1] Segment Anything Model. ICCV 2023.


**Experiments.** For Table 2, did the authors use other methods than K-means? The K-means approach appears to be almost equivalent to the proposed approach, although the proposed approach has been modified to achieve the best results, while the K-means approach I assume has not been.

Additionally, this work has been applied to CLIP, but CLIP is already fairly updated while there are several other existing methods that are better, such as BLIPv2, LLaVa, and others. We cannot be certain whether the problem of object localization properties still exists in more advanced methods, so I would like to see whether the approach can generalize well to other models.

**Questions:**

I am concerned that this paper does not present a significant approach to one specific VL model, such as CLIP, for dealing with object localization in a training-free manner. My concerns have been listed above, and I would appreciate it if the authors could address them. I am also open to the authors' feedback and other reviewers' opinions.

---

> ### Author Response · Authors · 2023-11-13
> **Response to  uWg6**
>
> Thank you for reviewing our paper, pointing out the strengths of our work, and providing valuable feedback!
> We hope that we can address some concerns in the following:
>
> ## Weaknesses
>
> ### Novelty
> GEM vs. CLIPSurgery(CS): The authors of CS concluded based on initial visualization that "the parameters key and query in self-attention link features from opposite semantic regions" (https://arxiv.org/pdf/2304.05653.pdf, Sec. 3.2, page 5), thus that key and query attention do break the semantic alignment between visual features and text. While this might be true for regular k-q attention, we argue that this is not the case if key and query projection are used in a self-self attention fashion (e.g., shown in Tab 1). So, depending on the dataset, key-key or query-query attention alone can even outperform value-value attention, thus having similar alignment capabilities as the value projection. We see this as a valuable insight that goes beyond the original CS paper.
> Further, as there seems to be no clear 'winner' in case of what performs best, k-k, q-q, v-v attention (also depending on the dataset), we further propose to ensemble all three representations, which, especially after normalization, helps to achieve consistently good performance across a variety of datasets without any need to fine-tune or test for a specific setting (note that optimizing e.g., by selecting the best out of k-k, q-q, v-v attention would also require some GT annotation for each separate use case).
>
> ### Motivation
> We think that a training-free localization pipeline can thought of more as a complement to semantic region or bounding box prediction methods, instead of a direct competition. While the proposed method provides a good region-specific vision-language alignment, but doe not capture e.g. crisp outlines, mask or region proposal methods are good in capturing those image properties, but need the vision-language alignment as a postprocessing step.
>
> We hope that GEM can help here: First, for mask proposal methods the regions need to be defined before the prompting, meaning there are limited ways to decompose, e.g., dog and dog collar, resp. the decomposition would have to be defined before the prompting and the segmentation needs to be adapted, which would raise a lot of practical problems (mainly meaning that we would have to tune hyper-parameters like granularity for each set of prompts separately). The advantage of a method like GEM could be that it is possible to overcluster an image and use the less sharp, but better aligned regions to choose the relevant segments.
>
> We assume that this could benefit both sides as current post-hoc alignment (e.g. for SAM) is usually done based on some other model like clip. This means for the alignment two options exist: 1) the ROI can be processed as a regular image by e.g., CLIP, which means that every single ROI needs to be forwarded to compute the VL alignment, which is very resource-intensive. The second option is to forward the full image but only use the tokens in the ROI to do the VL alignment. In this case, the model would have to rely on the last layer image token-text alignment, which is actually inverse to the actual prompt (see e.g. Cs and our results in Fig 3c) and therefore has only limited usability here.
>
> Based on you request, we run a comparative evaluation of GroudningSAM (https://arxiv.org/abs/2307.04767, https://github.com/IDEA-Research/Grounded-Segment-Anything, note that this is not published currently) which is not training-free, compared to GEM on OpenImageV7. GroundingSAM reaches a p_mIoU of 53.6 but with an average runtime of 1.18s per image, while GEM reaches 51.9 (with MetaCLIP backbone) and a runtime of 0.028s per image. We assume that the slight advantage of SAM over GEM is based on sharper masks. We will work on an implementation of merging SAM and GEM for the rebuttal deadline or the camera-ready version.

---

> ### Author Response · Authors · 2023-11-13
>
> ### Experiments.
> *Other methods than K-means?:* Thanks for the hint, we so far did not consider other clustering approaches beyond K-means. While we agree that the K-means approach performs almost comparable to the proposed approach, we have to point out that here, not the proposed approach (GEM) has been modified/finetuned to achieve the best results (best results at 1-2 iterations (45.5/46.2 for VOC and 32.6/31.9 for Context), but it is actually the K-means approach that needs an appropriate hyper-parameter tuning per dataset to perform comparable (best results at 3 iters for VOC (44.1), 7 iter for Context (31.1)). Note that, to tune this for each dataset, we would again need GT annotation, while GEM with 2 iterations provides good (not-optimized) results across all tested dataset.
>
> *Other backbones beyond CLIP:* We indeed extended our work in the meantime to more backbones with currently best results for metaclip400m (46.8 on VOC, 34.5 on Context, 51.9 on OV7). We will extend the Table in the paper. An extension to the two mentioned architecture might be hard to get working out of the box as BLIPv2 proposes a Q-Former module on to of a frozen image backbone (BLIP) and adapts the output of this backbone. It is therefore fair to assume that GEM features would not align with the pretrained BLIPv2 Q-Former and therefore require retraining which would be not feasible for us in the time of the rebuttal. We would therefore refer here to the results of the original BLIP model in Tab. 4(as the image backbone is frozen in BLIPv2, the representation should not change here). LLaVa follows the same reasoning, also building on a frozen image model and adapting the language pipeline to its output, only that here a CLIP ViT-L/14 is used, also evaluated in CLIP ViT-L/14. Overall, we agree that connecting the proposed architecture with higher-level language-image instruction-following approaches is an interesting and exciting direction, but we assume that it might require some amount of additional effort and might be out of the scope of this rebuttal.
>
>
> ## Questions.
> *I am concerned that this paper does not present a significant approach to one specific VL model, such as CLIP, for dealing with object localization in a training-free manner.*
>
> We agree that the proposed paper presents a technique that is generally applicable for a broad range of transformer-based VL models, but we don't see this as a negative point. Could you probably elaborate how having an approach for one specific VL model could be more advantageous here?
>
>
> Please let us know if you have any other questions or concerns regarding our paper or response.